# Study of the Relationship between Urban Expansion and PM$_{10}$ Concentration Using Multi-Temporal Spatial Datasets and the Machine Learning Technique: Case Study for Daegu, South Korea

**Yun-Jae Choung [1,2] and Jin-Man Kim [3,*]**

[1]  GIS Research Center, Geo C&I Co., Ltd., Daegu 41165, Korea; choung12osu@gmail.com
[2]  Global Land Satellite Information Center, Kyungpook National University, Daegu 41566, Korea
[3]  Geotechnical Engineering Research Institute, Korea Institute of Civil Engineering and Building Technology, Goyang-si, Gyeonggi-do 10223, Korea
*   Correspondence: jmkim@kict.re.kr; Tel.: +82-31-910-0221

**Abstract:** To protect the population from respiratory diseases and to prevent the damages due to air pollution, the main cause of air pollution should be identified. This research assessed the relationship between the airborne particulate concentrations (PM$_{10}$) and the urban expansion in Daegu City in South Korea from 2007 to 2017 using multi-temporal spatial datasets (Landsat images, measured PM$_{10}$ data) and the machine learning technique in the following steps. First, the expanded urban areas were detected from the multiple Landsat images using support vector machine (SVM), a widely used machine learning technique. Next, the annual PM$_{10}$ concentrations were calculated using the long-term measured PM$_{10}$ data. Finally, the degrees of increase of the expanded urban areas and of the PM$_{10}$ concentrations in Daegu from 2007 to 2017 were calculated by counting the pixels representing the expanded urban areas and computing variation of the annual PM$_{10}$ concentrations, respectively. The experiment results showed that there is a minimal or even no relationship at all between the urban expansion and the PM$_{10}$ concentrations because the urban areas expanded by 55.27 km$^2$ but the annual PM$_{10}$ concentrations decreased by 17.37 μg/m$^3$ in Daegu from 2007 to 2017.

**Keywords:** coarse particle; particulate matter 10 (PM$_{10}$); landsat image; machine learning; support vector machine

## 1. Introduction

Urban expansion, also called "urban sprawl," is defined as "the spreading of urban development (e.g., houses, shopping centers) on undeveloped lands near a city" or "the rapid expansion of the geographic extent of cities and towns, often characterized by low-density residential housing, single-use zoning, and increased reliance on private automobiles for transportation" [1,2]. In general, urban expansion has a close relationship with urban development, infrastructure improvement, population growth, etc. [3].

Coarse particle, defined as particulate matter 10 (PM$_{10}$), consists of particles with a diameter of 10 μm or less [4,5]. PM$_{10}$ is one of the main components of air pollution, and it also results in various environmental impacts (e.g., atmospheric pollution) and human health impacts (e.g., chronic respiratory diseases) [6,7]. In particular, exposure to a high PM$_{10}$ concentration can cause a number of significant health impacts, ranging from coughing to high blood pressure, heart attack, stroke, and lung cancer [8].

Previous studies found that urban expansion generally has a significant impact on air pollution because more human activities that can cause air pollution (e.g., vehicular traffic) are expected in urban areas [9]. Stone (2007) assessed the relationship between urban expansion and air quality [10]. Cho and Choi (2014) investigated the effect of compact urban development on air quality [11]. Liu et al. (2018) assessed the relationship between urban air pollution and urban form, seasonality, and city size [12].

To protect the population from respiratory diseases and to prevent public health disasters due to $PM_{10}$ concentration, the main causes of $PM_{10}$ concentration in the city should be identified. Limited research has been conducted, however, to identify the main causes of $PM_{10}$ concentration in each city in South Korea. In general, urban expansion is regarded as the main cause of urban air pollution owing to the urban development activities accompanying it. This research aims to assess the relationship between urban expansion and $PM_{10}$ concentration by monitoring the 10 years (from 2007 to 2017) of urban expansion and the annual $PM_{10}$ concentrations using multi-temporal spatial datasets acquired in Daegu, South Korea and the machine learning technique. First, the expanded urban areas were detected from the multi-temporal Landsat satellite images using the machine learning technique. Then the annual PM10 concentrations were calculated using the long-term measured PM10 data. Finally, the relationship between urban expansion and $PM_{10}$ concentration was assessed by counting the pixels representing the expanded urban areas and computing variation of the annual PM10 concentrations, respectively.

## 2. Study Area and Datasets

Daegu Metropolitan City in South Korea was selected as the study area in this research for the following reasons. First, the urban areas of Daegu have been significantly expanded of late [13]. Second, there are long-term measured coarse particle ($PM_{10}$) data acquired by the 11 air quality monitoring stations (AQMSs) in Daegu, which can be used for the study [14]. Daegu has operated these AQMSs since 1973 for sustainably monitoring the air quality condition of the city. Figure 1 shows the locations of the 11 AQMSs in Daegu, South Korea.

The multi-temporal Landsat satellite images acquired on 13 May 2007 ("first Landsat image") and on April 29, 2017 ("second Landsat image") were used in this study for the following reasons. First, the urban areas in Daegu significantly expanded during such periods due to the city's urban development policy. Second, both Landsat images were less affected by the prevailing atmospheric conditions then. Figure 2 shows one section each of the first and second Landsat images.

The first Landsat image was acquired by the Landsat-5 thematic mapper (TM) sensor, and it consists of seven bands (blue: 450–520 nm; green: 520–600 nm; red: 630–690 nm; near-infrared: 770–900 nm; short-wave infrared 1: 1550–1750 nm; short-wave infrared 2: 2080–2350 nm; and thermal: 10,400–12,500 nm) [15]. The second Landsat image was acquired by the Landsat-8 operational land imager (OLI) and the thermal infrared sensor (TIRS), and it consists of nine bands (coastal aerosol: 435–451 nm; blue: 452–512 nm; green: 533–590 nm; red: 636–673 nm; near-infrared: 851–879 nm; short-wave infrared 1: 1566–1651 nm; short-wave infrared 2: 2107–2294 nm; thermal infrared 1: 10,600–11,190 nm; and thermal infrared 2: 11,500–12,510 nm) [15]. Both Landsat images were georeferenced to the coordinate system universal transverse mercator (UTM), zone 52 N, based on the 1984 datum world geodetic system (WGS).

To measure the annual $PM_{10}$ concentrations during the same period for the monitoring of the urban expansion in Daegu using the Landsat 1 and 2 images, the measured $PM_{10}$ data between 2007 and 2017 acquired through the Daegu atmospheric information system (https://air.daegu.go.kr/) were also utilized.

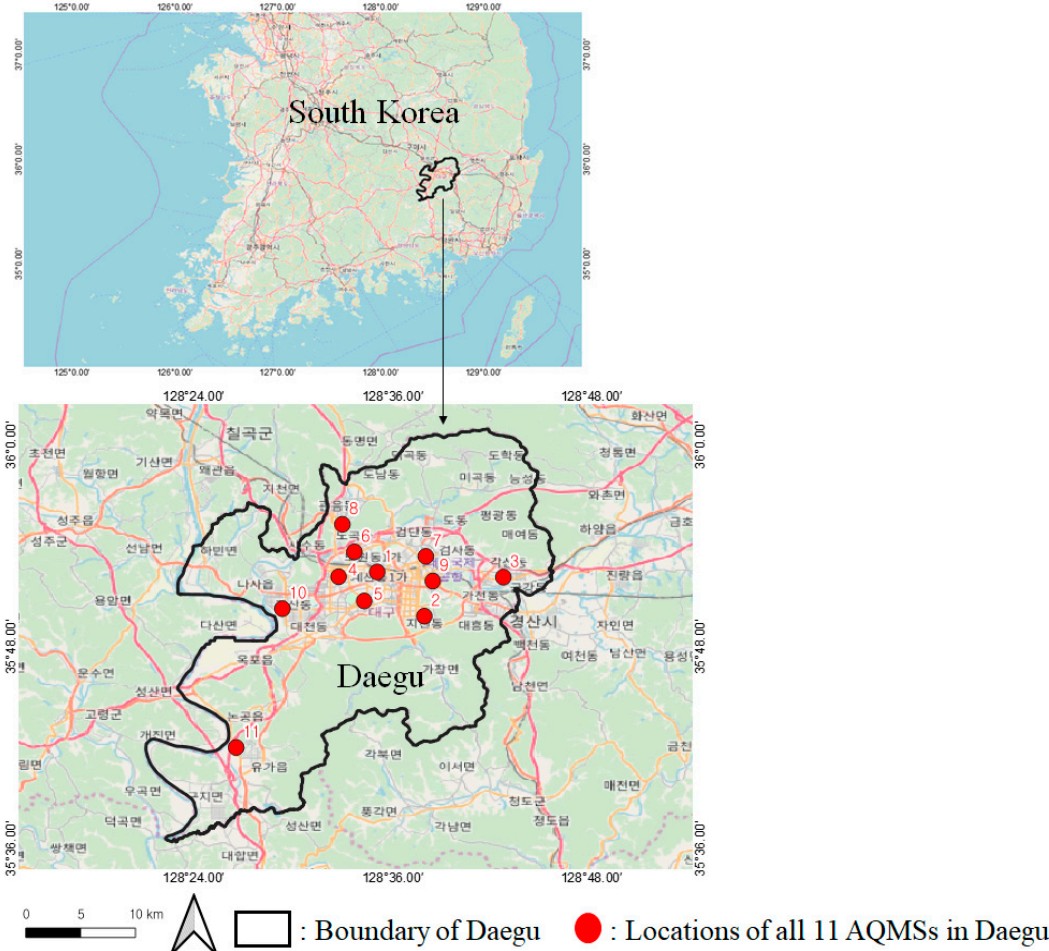

**Figure 1.** Locations of the 11 air quality monitoring stations (AQMSs) in Daegu, South Korea.

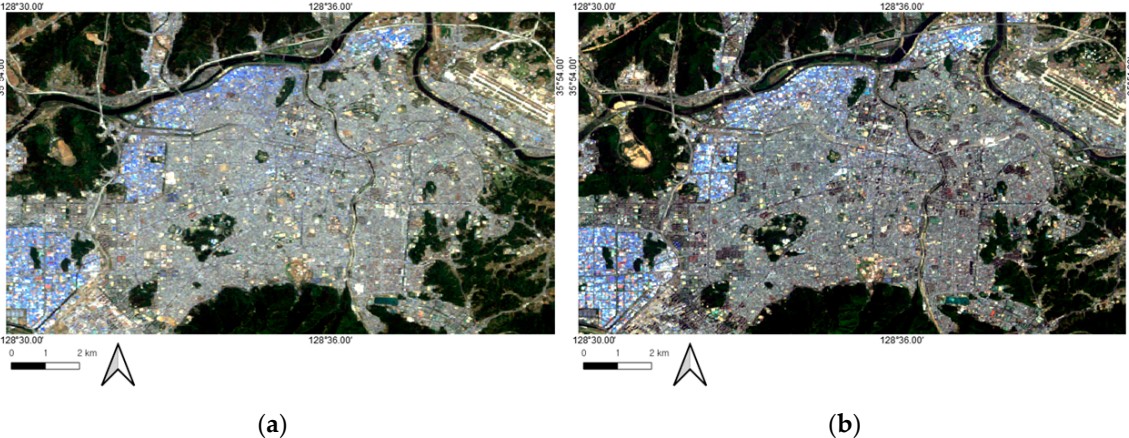

(**a**)                                                                                     (**b**)

**Figure 2.** One section each of the multi-temporal Landsat images utilized in this research: (**a**) a section of the first Landsat image (acquired on 13 May 2007); and (**b**) a section of the second Landsat image (acquired on 29 April 2017).

## 3. Methodology

Figure 3 presents a flowchart of the procedure that was employed to assess the relationship between urban expansion and $PM_{10}$ concentration in Daegu from 2007 to 2017 using the given datasets.

As can be seen in Figure 3, in the first step of the proposed methodology, two urban maps were generated, respectively, from the first and second Landsat images, using the support vector machine

(SVM) technique, a widely used machine learning technique. Then the extent of urban expansion was detected using the generated first and second urban maps. In the next step, the annual $PM_{10}$ concentrations were calculated using the measured $PM_{10}$ data acquired by each AQMS of Daegu. Finally, the relationship between the urban expansion and the $PM_{10}$ concentration rate in Daegu was assessed using the two calculated statistics: the increase of the expanded urban areas and the increase of the annual $PM_{10}$ concentrations in Daegu from 2007 to 2017.

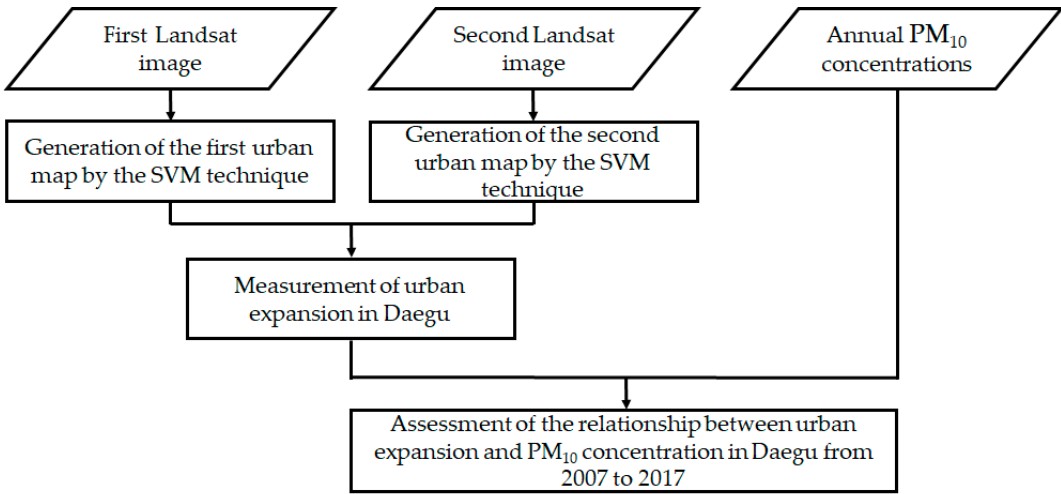

**Figure 3.** Flowchart showing the procedure that was employed to assess the relationship between urban expansion and $PM_{10}$ concentration in Daegu from 2007 to 2017 using the given datasets.

### 3.1. Generation of the Urban Maps by the SVM Technique

This section illustrates the procedure for generating urban maps through the SVM technique, a widely used machine learning algorithm. Machine learning is defined as ""the ability of a machine to improve its performance based on previous results" [16]. The machine learning technique has been widely used of late in remote sensing applications for classifying land uses and detecting the significant features from the remote sensing datasets, due to its advantages for high-value classification [17,18]. SVM, a widely used machine learning technique for finding the linear hyperplane that maximizes the margins between the two clusters in n-dimensional spaces, has been widely used in remote sensing applications due to its superior advantages over the other machine learning techniques for classifying land uses, detecting significant features, and avoiding classification errors [19]. Hence, in this research, the SVM technique was used to generate urban maps, which distinguish the urban areas from the non-urban areas (water, soil, vegetation, etc.). Figure 4 shows the first and second urban maps separately generated from the first and second Landsat images, respectively, through the SVM technique.

### 3.2. Detection of the Expanded Urban Areas in Daegu from 2007 to 2017

In the next step of the proposed methodology, the expanded urban areas in Daegu were detected using the two urban maps that had been generated. The expanded urban areas from 2007 to 2017 were detected by intersecting the pixels representing the non-urban areas in the first urban map and the pixels representing the urban areas in the second urban map. Figure 5 shows the locations of the AQMSs and expanded urban areas in Daegu from 2007 to 2017 detected using the two generated urban maps.

As can be seen in Figure 5, all the AQMSs in Daegu were located near the expanded urban areas detected using the two generated urban maps.

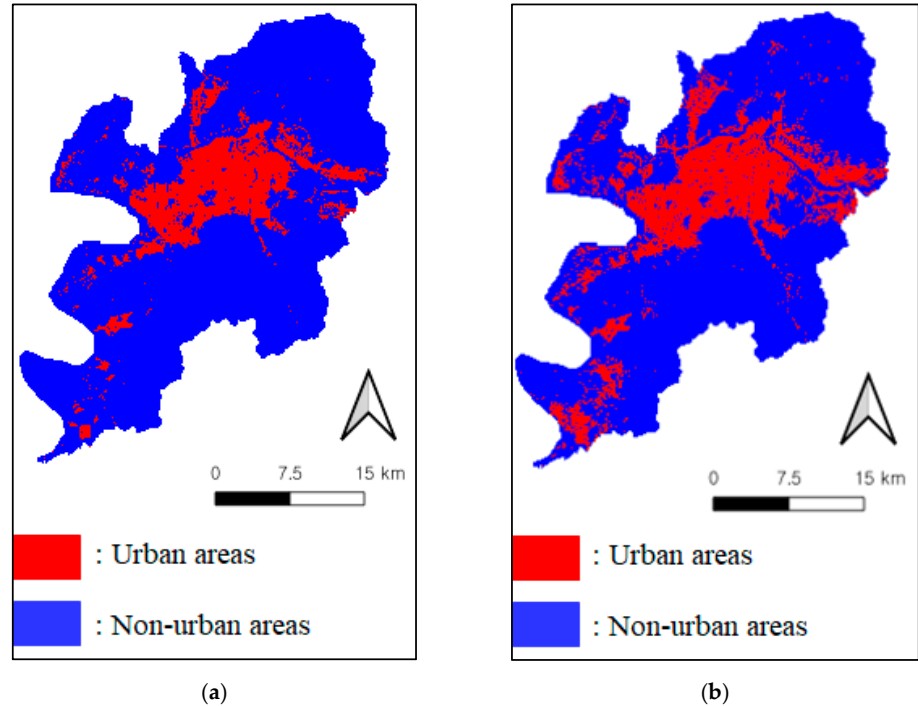

(**a**)                                                    (**b**)

**Figure 4.** First and second urban maps: (**a**) first urban map generated from the first Landsat image through the SVM technique; and (**b**) second urban map generated from the second Landsat image through the SVM technique.

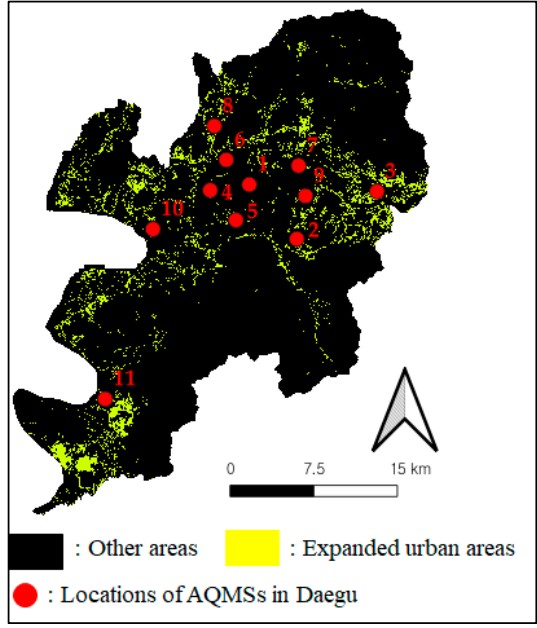

**Figure 5.** Locations of the AQMSs and expanded urban areas in Daegu from 2007 to 2017 detected using the two generated urban maps.

*3.3. Calculation of the Statistics for the Annual PM$_{10}$ Concentrations*

In this section, the calculation of the statistics for the annual PM$_{10}$ concentrations in each AQMS in Daegu from 2007 to 2017 is described. Figure 6 presents time series graphs showing the annual PM$_{10}$ concentrations in each AQMS in Daegu from 2007 to 2017.

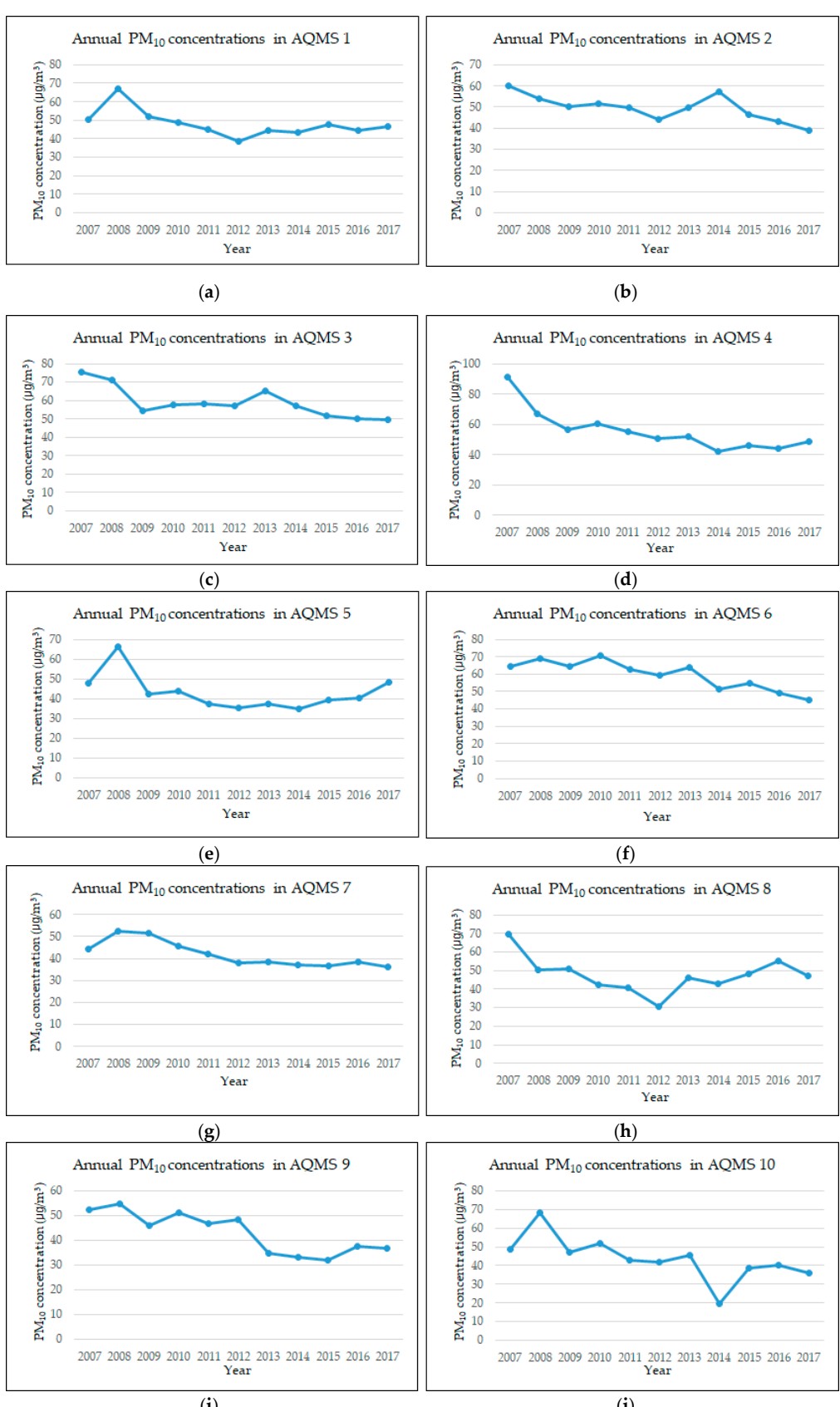

**Figure 6.** *Cont.*

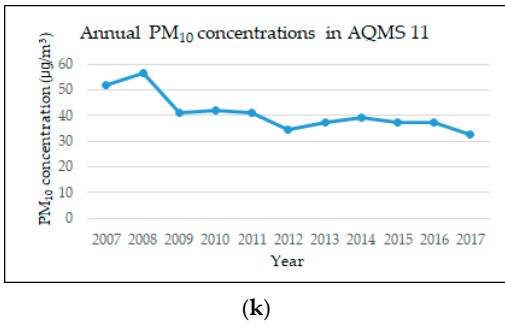

(k)

**Figure 6.** Time series graphs showing the annual $PM_{10}$ concentrations in each AQMS in Daegu from 2007 to 2017. AQMS 1(**a**) to AQMS 11(**k**).

## 4. Results and Discussions

### 4.1. Accuracies of the Generated Urban Maps

In this section, the degrees of accuracy of the first and second urban maps separately generated from the first and second Landsat images, respectively, are assessed using the 100 checkpoints generated through manual digitization. Figure 7 shows examples of the checkpoints generated through manual digitization for measuring the degrees of accuracy of the first and second urban maps.

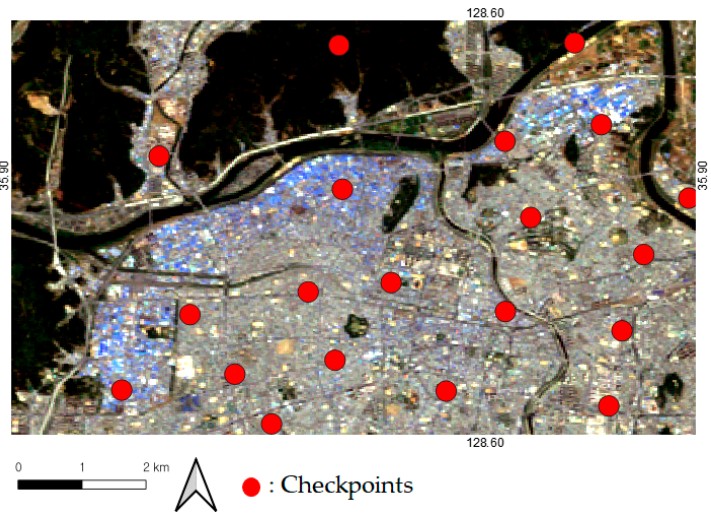

**Figure 7.** Examples of the checkpoints generated through manual digitization for measuring the degrees of accuracy of the first and second urban maps.

Table 1 shows the statistical results showing the degrees of accuracy of the first and second urban maps separately generated from the first and second Landsat images, respectively.

As can be seen in Table 1, the first and second urban maps separately generated from the first and second Landsat images, respectively, through the SVM technique had high accuracy in identifying the urban areas in the entire Daegu area. There were a few misclassification errors, however, in both urban maps because some urban features (e.g., man-made features) were misclassified as non-urban features (e.g., soil, water, vegetation), or vice versa. These misclassification errors were generally caused by the similar reflectance characteristics of the different materials owing to the shades, etc. [18].

**Table 1.** Statistical results showing the degrees of accuracy of the first and second urban maps separately generated from the first and second Landsat images, respectively: (**a**) degree of accuracy of the first urban map generated from the first Landsat image; and (**b**) degree of accuracy of the second urban map generated from the second Landsat image.

| (a) | | | |
|---|---|---|---|
| **Overall Accuracy** | | **97%** | |
| **Producer's Accuracy (Error of Omission)** | | **User's Accuracy (Error of Commission)** | |
| Urban areas | 94% | Urban areas | 100% |
| Non-urban areas | 100% | Non-urban areas | 94% |
| (b) | | | |
| **Overall Accuracy** | | **99%** | |
| **Producer's Accuracy (Error of Omission)** | | **User's Accuracy (Error of Commission)** | |
| Urban areas | 100% | Urban areas | 98% |
| Non-urban areas | 98% | Non-urban areas | 100% |

*4.2. Relationship between the Urban Expansions and the PM$_{10}$ Concentrations in Daegu from 2007 to 2017*

Discussed in this section is the relationship between the urban expansion and the annual PM$_{10}$ concentration rate in Daegu from 2007 to 2017 determined by calculating the following statistics: those showing the increase of the expanded urban areas in Daegu from 2007 to 2017 and those showing the annual PM$_{10}$ concentration changes in each AQMS, each year and each season in Daegu from 2007 to 2017 (see Table 2).

As can be seen in Figure 6, Table 2a,b, the urban areas expanded by 55.27 km$^2$ in the entire Daegu area from 2007 to 2017, but the annual PM$_{10}$ concentrations in all the AQMSs in Daegu only slightly increased by 0.45 μg/m$^3$ in only one station or sharply decreased by 4.12~42.28 μg/m$^3$ in the other 10 stations, within the same period. Table 2c shows that the annual PM$_{10}$ concentrations had decreased in Daegu City from 2007 to 2017 by 17.37 μg/m$^3$.

Table 2b,c show that the annual PM$_{10}$ concentrations were measured high in AQMSs 3, 4, and 6, while they were measured low in AQMSs 5, 7, 9, 10 and 11. Table 2c also shows that the highest annual PM$_{10}$ concentrations was most frequently measured in AQMS 6 from 2007 to 2017 while the lowest annual PM$_{10}$ concentrations most frequently measured in AQMS 11 during the same period. Based on Table 2b,c results, we assume that there are the number of other facilities (e.g., industrial factories) that emit the PM$_{10}$ particles near AQMSs 3, 4, and 6, while there are few facilities emitting the PM$_{10}$ particles near AQMS 5, 7, 9, 10, and 11.

Table 2c,d show that the annual PM$_{10}$ concentrations generally measured high in spring and winter compared to summer and autumn. In addition, the annual PM$_{10}$ concentrations decreased in all the seasons by 31.11 μg/m$^3$ for spring, by 8 μg/m$^3$ for summer, by 8.78 μg/m$^3$ for autumn, and by 18.77 μg/m$^3$ for winter. Based on Table 2d, we assume that the climate factors (e.g., air temperature, air pressure, rainfall, humidity, wind speed, and wind direction) can be significant for the annual PM$_{10}$ concentrations.

The above experiment results show that there is a minimal or no relationship at all between the urban expansion and the PM$_{10}$ concentrations rate in Daegu, which means that the urban expansion that occurred in Daegu from 2007 to 2017 was not the main cause of the rise in the PM$_{10}$ concentration rate in Daegu, South Korea during the same period. We assume that, however, the types of facilities and climate factors can influence on the annual PM$_{10}$ concentrations.

**Table 2.** Statistics showing the increase of the expanded urban areas in Daegu from 2007 to 2017 and the annual $PM_{10}$ concentration changes in each air quality monitoring stations (AQMS) in Daegu within the same period: (**a**) statistics showing the total number of urbanized areas in Daegu from 2007 to 2017; (**b**) statistics showing the annual $PM_{10}$ concentration changes in each AQMS in Daegu from 2007 to 2017; (**c**) statistics showing the range of variability of the annual $PM_{10}$ concentrations in each year from 2007 to 2017; and (**d**) statistics showing the annual $PM_{10}$ concentrations in each season (spring: March, April, and May; summer: June, July, and August; autumn: September, October, and November; and winter: December, January, and February) from 2007 to 2017.

| (a) | | |
|---|---|---|
| Total Areas of the Urban Areas in the First Urban Map ($km^2$) | Total Areas of the Urban Areas in the Second Urban Map ($km^2$) | Increase of the Expanded Urban Areas in Daegu from 2007 to 2017 ($km^2$) |
| 148.08 | 203.35 | + 55.27 |

| (b) | | | | | |
|---|---|---|---|---|---|
| AQMS ID | Maximum ($\mu g/m^3$) | Minimum ($\mu g/m^3$) | Average ($\mu g/m^3$) | Standard Deviation | Variation of Annual $PM_{10}$ Concentration (2017 vs 2007) ($\mu g/m^3$) |
| AQMS 1 | 67.28 | 38.79 | 48.07 | 7.35 | −4.12 |
| AQMS 2 | 60.21 | 38.80 | 49.61 | 6.30 | −21.41 |
| AQMS 3 | 75.32 | 49.30 | 58.80 | 8.40 | −26.02 |
| AQMS 4 | 91.14 | 41.99 | 55.85 | 13.79 | −42.28 |
| AQMS 5 | 66.16 | 34.64 | 42.95 | 8.97 | +0.45 |
| AQMS 6 | 70.65 | 45.23 | 59.54 | 8.34 | −19.29 |
| AQMS 7 | 52.22 | 36.29 | 41.85 | 5.78 | −7.88 |
| AQMS 8 | 56.75 | 32.78 | 47.61 | 9.72 | −22.75 |
| AQMS 9 | 54.86 | 31.78 | 43.03 | 8.43 | −15.59 |
| AQMS 10 | 68.53 | 19.66 | 43.78 | 11.85 | -12.81 |
| AQMS 11 | 56.75 | 32.78 | 41.05 | 7.26 | −19.35 |

| (c) | | | | | | |
|---|---|---|---|---|---|---|
| Year | Maximum ($\mu g/m^3$) | AQMS ID for Maximum | Minimum ($\mu g/m^3$) | AQMS ID for Minimum | Average ($\mu g/m^3$) | Standard Deviation |
| 2007 | 91.14 | AQMS 4 | 44.17 | AQMS 7 | 59.72 | 14.31 |
| 2008 | 71.04 | AQMS 3 | 50.36 | AQMS 8 | 61.58 | 7.83 |
| 2009 | 64.37 | AQMS 6 | 41.21 | AQMS 11 | 50.52 | 6.59 |
| 2010 | 70.65 | AQMS 6 | 42.27 | AQMS 11 | 50.51 | 8.66 |
| 2011 | 62.63 | AQMS 6 | 37.21 | AQMS 5 | 47.42 | 8.15 |
| 2012 | 59.51 | AQMS 6 | 30.74 | AQMS 8 | 43.61 | 9.44 |
| 2013 | 65.03 | AQMS 3 | 34.63 | AQMS 9 | 46.73 | 10.34 |
| 2014 | 57.22 | AQMS 2 | 19.66 | AQMS 10 | 41.63 | 11.02 |
| 2015 | 54.55 | AQMS 6 | 31.78 | AQMS 9 | 43.48 | 7.13 |
| 2016 | 54.98 | AQMS 8 | 31.32 | AQMS 11 | 43.6 | 5.71 |
| 2017 | 49.30 | AQMS 3 | 32.78 | AQMS 11 | 42.35 | 6.21 |

| (d) | | | | |
|---|---|---|---|---|
| Year | Spring ($\mu g/m^3$) | Summer ($\mu g/m^3$) | Autumn ($\mu g/m^3$) | Winter ($\mu g/m^3$) |
| 2007 | 81.58 | 41.55 | 48.47 | 64.32 |
| 2008 | 73.63 | 52.11 | 55.88 | 64.10 |
| 2009 | 51.74 | 42.37 | 45.51 | 62.87 |
| 2010 | 57.39 | 39.86 | 50.21 | 58.59 |
| 2011 | 59.77 | 37.59 | 42.20 | 50.59 |
| 2012 | 50.63 | 33.98 | 41.76 | 48.14 |
| 2013 | 52.92 | 38.99 | 39.90 | 55.02 |
| 2014 | 50.01 | 33.82 | 35.30 | 47.49 |
| 2015 | 50.51 | 34.59 | 31.14 | 57.61 |
| 2016 | 53.33 | 32.71 | 40.02 | 48.49 |
| 2017 | 50.47 | 33.55 | 39.69 | 45.55 |

## 5. Conclusions and Future Works

In this research, an experiment was performed to assess the relationship between the urban expansion and the $PM_{10}$ concentration rate in Daegu from 2007 to 2017 by calculating the expanded urban areas and the annual $PM_{10}$ concentration changes in each AQMS, each year and each season. The experiment results showed that there is a minimal or no relationship at all between the urban expansion that occurred in Daegu from 2007 to 2017 and the rise in the $PM_{10}$ concentration rate in the

same city during the same period because the urban areas significantly expanded but the annual $PM_{10}$ concentrations sharply decreased.

This research proved that the urban expansion that occurred in Daegu from 2007 to 2017 was not the main cause of the rise in the $PM_{10}$ concentration rate in Daegu during the same period. This research, however, also suggested that the other factors, such as types of facilities or climate factors, can be significant for the annual $PM_{10}$ concentrations. To protect the public health, it is necessary to identify the main causes of $PM_{10}$ concentrations, hence, further research, will be carried out to identify the main causes (e.g., the climate factors and the type of facilities) of $PM_{10}$ concentrations in general. In addition, research will be conducted to identify the main cause of $PM_{2.5}$ concentrations that also causing serious air pollution problems.

**Author Contributions:** Y.-J.C. performed the experiments, analyzed the data and wrote the manuscript; J.-M.K. proposed the method, designed the experiments and modified the manuscript.

**Funding:** This research was supported by a grant (18SCIP-B065985-06) from Smart Civil Infrastructure Research Program funded by Ministry of Land, Infrastructure and Transport (MOLIT) of Korea government and Korea Agency for Infrastructure Technology Advancement (KAIA).

**Conflicts of Interest:** The authors declare no conflict of interest.

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
