# Peer review of "Study of the Relationship between Urban Expansion and PM10 Concentration Using Multi-Temporal Spatial Datasets and the Machine Learning Technique: Case Study for Daegu, South Korea"

_applsci, doi:10.3390/app9061098_

Reviewer 1 Report

Please find comments in the attached file.

Author Response

Minor comments

1. (title) the title is not clear. I suggest to change it in “Study of the relationship between urban expansion and PM10 concentration using multi-temporal spatial datasets and the machine learning technique: case study for Daegu, South Korea”

(Answer) We appreciate your comment. We modified the manuscript.

2. (Through the manuscript) Please indicate PM10 using the subscript, as per convention (i.e., PM10)

(Answer) We appreciate your comment. We modified the manuscript.

3. (Through the manuscript) PM10 is conventionally classified as coarse particle and not as fine-dust or fine particle.

(Answer) We appreciate your comment. We modified the manuscript.

4. (Through the manuscript) Please refer to PM10 concentrations and not to PM10 density

(Answer) We appreciate your comment. We modified the manuscript.

5. (Through the manuscript) Please refer generally to PM10 concentrations and not to PM10 pollution (or fine-dust pollution, etc)

(Answer) We appreciate your comment. We modified the manuscript.

6. (abstract, line 15): please substitute “civilians” with “population”

(Answer) We appreciate your comment. We modified the manuscript.

7. (abstract, line 16-17): Please re-phrase as follows “This research assessed the relationship between the airborne particle concentrations (PM10) and the urban expansion in Daegu City in South Korea from 2007 to 2017…”

(Answer) We appreciate your comment. We modified the manuscript.

8. (abstract, line 20): Please write in full all acronyms the first time they appear in the text. In this case, define “SVM”

(Answer) We appreciate your comment. We modified the manuscript.

9. (abstract, line 22-23): “the expanded urban areas and of the PM10 22 density in Daegu from 2007 to 2017 were calculated” Please define how it was calculated

(Answer) We appreciate your comment. We modified the manuscript.

10. (abstract, line 25-26): “the urban areas significantly expanded but the annual PM10 density sharply decreased in Daegu 25 from 2007 to 2017” Please define how much the urban area has increased and how much the concentration of PM10 has decreased

(Answer) We appreciate your comment. We modified the manuscript.

11.(Introduction, line 37) PM10 is conventionally classified as coarse particle and not as fine-dust or fine particle.

(Answer) We appreciate your comment. We modified the manuscript.

12. (Introduction, line 37) Please delete “tiny” from “consist of tiny particles”

(Answer) We appreciate your comment. We modified the manuscript.

13. (Introduction, line 38) Please rephrase as follows “PM10 is one of the main components of air pollution,”

(Answer) We appreciate your comment. We modified the manuscript.

14. Figures 1 and 2 do not add anything useful to the manuscript. Please consider removing both the figures.

(Answer) We appreciate your comment. We modified the manuscript.

15. (Introduction, line 47) Please substitute “(e.g., car use)” with “(e.g., vehicular traffic)”

(Answer) We appreciate your comment. We modified the manuscript.

16. (Introduction, line 48-51) Please discuss briefly what emerges from these studies

(Answer) We appreciate your comment. We modified the manuscript.

17. (figure 4) I suppose that figure 4 a is the one on the left and the hail 4 b is the one on the left. Specify it for clarity in the caption

(Answer) We appreciate your comment. We clarified it in the manuscript.

18. (table 2a) what is the level of accuracy of the estimation of urban areas? ± 0.01Km2?

(Answer) We appreciate your comment. The level of accuracy of both urban areas is ± 0.01km2.

19.(table 2b) in the last column on the right I would indicate "variation of annual PM10 concentration (2017 vs 2007)”?

(Answer) We appreciate your comment. We indicated “variation of annual PM10 concentration (2017 vs 2007)” in the last column on the right in Table 2b.

20. (Figure 8). Please indicate the unit [µg/m3 ] on the x-axis of the graphs. I suggest also to indicate the range of variability (min-max? Standard deviation?) For each year. Finally, I suggest adding another graph, relative to the annual mean concentration (and standard deviation / or range min-max) averaged over the 11 AQMS. This last point also applies for table 2b

(Answer) We appreciate your comment. We revised the manuscript. And, we also added an additional table on Table 2(c) showing that the range of variability of the annual PM10 concentrations in each year from 2007 to 2017.

Major comments

1. (Methodology, line 105-108) Please Explain how the relationship between urban expansion and PM10 variation was analyzed. Which technique was used? If only a comparison was made between the increase in urban areas (+55.27 km2 ) and a reduction in the average annual value of the PM10 concentration (2017 - 2007), I would say that the conclusions obtained are not solid. particularly advanced techniques have been used to quantify the increase in urbanized area, but the subsequent analysis of the data seems to me rather elementary. There is no kind of statistical assumption. Are the changes in PM concentrations significant? What is the variability of PM10for each year? Any kind of possible co-variate and confounding factors (weather conditions, rainfall, traffic flows, changes in industrial emissions, just to list some) have not been considered in this analysis.

(Answer) We appreciate your comment. Considering your suggestions, we added the additional tables (Tables 2(c) and (d)) that shows the range of variability of the annual PM10 concentrations in each year and the annual PM10 concentrations in each season from 2007 to 2017. Based on the reinforced results (Tables 2(c) and (d)), we did additional analysis and had the improved conclusions. However, due to the limitation of data acquisition, the further research couldn’t be carried out. In the future, the further research (assessing the relationship between PM10 concentrations and other factors such as weather conditions/rainfall/traffic flows/industrial emission) would be carried out using the additional datasets.

2. (Results and Discussion – line 204-210) There is virtually no discussion of the results. And the analysis of the obtained results to investigate the PM10-urban area expansion relationship, is limited to a simple comparison. This is not enough to support any kind of conclusion.

(Answer) We appreciate your comment. As we mentioned in the above paragraph, we did additional analysis based on the added Tables 2(c) and (d) for supporting the improved conclusions.

3. (Conclusions and Future Works – line 212-221) These conclusions are not supported by the results obtained and presented, because these do not constitute proof of the absence of a relationship between the two phenomena under analysis. Please provide additional supporting data or use more robust and robust methods for data analysis.

(Answer) We appreciate your comment. Again, we did additional analysis based on the added Tables 2(c) and (d) for supporting the improved conclusions.

4. (Conclusions and Future Works – line 222-223) please consider these comments also in the imposition of future studies.

(Answer) We appreciate your comment. We modified the conclusion section in the manuscript.

Reviewer 2 Report

major comments:

1.) The conclusion of this study that despite significant urban expansion, the PM10 concentrations have declined is indeed very interesting. Have the authors looked at meteorological changes during this period. For example, changes in precipitation over this region? 

2.) The authors only show changes in annual means across this period. What about the changes in different seasons? Do they show a similar decline? It would be helpful to look at changes in three monthly averages. For example, winter (December,January and February average), summer (May, June, July average).

3.) In Figure 7, I cannot help but notice that most of the stations are located on the fringes of the expanded urban areas (i.e. in black areas)[e.g. station 11, station 10, station 1]. Only stations 3 and 8 look like they are in the middle of the urban expansion areas. I feel that the results could be different if we look at measurements located closer to the expanded areas. Do the authors agree with this?

If so, are there additional measurement datasets available closer to the expanded areas?

4.) Introduction: "This research aimed to identify the main cause of fine-dust pollution by assessing the relationship ". I don't think the authors identified the main cause. They just eliminated urban pollution as a potential cause.

5.) What about the trends in PM2.5? Can the authors show trends in PM2.5 over the past few years in this city. Not necessarily 11 stations but even having one or two stations should be enough.

minor comments:

1.) line 46: replace "Previous researches found" with "Previous studies found".

2.) line 56: replace "This research aimed" with "This research aims".

3.) Figure 8 caption is unnecessarily lengthy, shorten it. E.g.  Time series graphs showing the annual PM10 density in each AQMS in Daegu from 2007 to 2017. AQMS 1 (a) to AQMS 11 (k).

Author Response

Major comments:

1.) The conclusion of this study that despite significant urban expansion, the PM10 concentrations have declined is indeed very interesting. Have the authors looked at meteorological changes during this period? For example, changes in precipitation over this region?

(Answer) We appreciate your comment. We are also very interested in the relationship between the meteorological changes and the fine dust concentrations. However, this study was designed to only focus on the relationship between the PM 10 concentrations and the urban expansions. Hence, in future study, we will focus on the relationship between the fine dust concentrations and the meteorological changes and mention it in the conclusion section of the revised manuscript.

2.) The authors only show changes in annual means across this period. What about the changes in different seasons? Do they show a similar decline? It would be helpful to look at changes in three monthly averages. For example, winter (December, January and February average), summer (May, June, July average).

(Answer) We appreciate your comment. We added the statistics showing the average annual PM10 concentrations in the different seasons in Table 2(d).

3.) In Figure 7, I cannot help but notice that most of the stations are located on the fringes of the expanded urban areas (i.e. in black areas)[e.g. station 11, station 10, station 1]. Only stations 3 and 8 look like they are in the middle of the urban expansion areas. I feel that the results could be different if we look at measurements located closer to the expanded areas. Do the authors agree with this?

If so, are there additional measurement datasets available closer to the expanded areas?

(Answer) We appreciate your comment. First of all, there is no additional measurement dataset, which means that the additional experiments currently cannot be carried out. And, as seen in Figure 6, the annual PM10 concentrations has significantly or slightly decreased in all the 11 stations during the same period regardless of the distances between the stations and the expanded urban areas. Hence, in my opinion, the distance between the station and the expanded urban area might not be the main factor for monitoring the annual PM10 concentration, which means that the results might not be different although we look at the additional measurements located closer to the expanded areas.

4.) Introduction: "This research aimed to identify the main cause of fine-dust pollution by assessing the relationship ". I don't think the authors identified the main cause. They just eliminated urban pollution as a potential cause.

(Answer) We appreciate your comment, and we totally agreed with your comment. Hence, we modified replace the previous sentence in the manuscript with the new sentence “This research aims to assess the relationship between urban expansion and PM10 concentration by monitoring the 10 years of urban expansions and the annual PM10 concentrations using multi-temporal spatial datasets and the machine learning technique.”

5.) What about the trends in PM2.5? Can the authors show trends in PM2.5 over the past few years in this city. Not necessarily 11 stations but even having one or two stations should be enough.

(Answer) We appreciate your comment. Unfortunately, the PM 2.5 data is not available in Daegu city. We also agree that, however, the annual PM 2.5 concentration is very important factor for measuring air pollution level. In the future study, hence, we will use the PM 2.5 data acquired in other cities of South Korea for measuring air pollution level in the study area.

Minor comments:

1.) line 46: replace "Previous researches found" with "Previous studies found".

(Answer) We appreciate your comment. We modified the manuscript.

2.) line 56: replace "This research aimed" with "This research aims".

(Answer) We appreciate your comment. We modified the manuscript.

3.) Figure 8 caption is unnecessarily lengthy, shorten it. E.g.  Time series graphs showing the annual PM10 density in each AQMS in Daegu from 2007 to 2017. AQMS 1 (a) to AQMS 11 (k).

(Answer) We appreciate your comment. We modified the manuscript.

Round  2

Reviewer 1 Report

The revised version of the manuscript is better than the original version. In my opinion, the study still presents some methodological and performance-related limitations, but these are now presented and discussed in the text.

Some minor comments (number 2, 3, 11 and 16) have not been properly replied. Even if minor, these issues should still be considered before evaluating the publication of the manuscript.

Author Response

The revised version of the manuscript is better than the original version. In my opinion, the study still presents some methodological and performance-related limitations, but these are now presented and discussed in the text.

Some minor comments (number 2, 3, 11 and 16) have not been properly replied. Even if minor, these issues should still be considered before evaluating the publication of the manuscript.

(Answer) We appreciate your comments. We modified the manuscript following your previous comments (2, 3, 11 and 16) illustrated below.

Comment 2) (Through the manuscript) Please indicate PM10 using the subscript, as per convention (i.e., PM10)

(Answer) We appreciate your comment. We modified the manuscript (replaced PM10 with PM10).

Comment 3) (Through the manuscript) PM10 is conventionally classified as coarse particle and not as fine-dust or fine particle.

(Answer) We appreciate your comment. We replaced with fine dust with coarse particle in the entire manuscript.

Comment 11) (Introduction, line 37) PM10 is conventionally classified as coarse particle and not as fine-dust or fine particle.

(Answer) We appreciate your comment. We replaced fine dust with coarse particle in the introduction, line 37.

Comment 16) (Introduction, line 48-51) Please discuss briefly what emerges from these studies

(Answer) We appreciate your comment. We discussed briefly what emerges from these studies in the end of the introduction section.

Reviewer 2 Report

Thanks for addressing my comments and improving the manuscript. It looks much better now. Your findings were very interesting and I hope you will follow up with a paper indicating the cause of PM10 declines. I do not have any more comments.

Author Response

Thanks for addressing my comments and improving the manuscript. It looks much better now. Your findings were very interesting and I hope you will follow up with a paper indicating the cause of PM10 declines. I do not have any more comments.

(Answer) We appreciate your comments. We are going to continue this research for finding the more valuable results. Thank you.
